# OFA³: Automatic Selection of the Best Non-dominated Sub-networks for Ensembles

**Anonymous**[1]

[1]Anonymous Institution

**Abstract**  Advancement of Neural Architecture Search (NAS) has the potential to significantly improve the efficiency and performance of machine learning systems, as well as enable the exploration of new architectures and applications across a wide range of fields. A promising direction for developing more scalable and adaptive neural network architectures is the Once-for-All (OFA), a NAS framework that decouples the training and the search stages, meaning that one super-network is trained once, and then multiple searches can be performed according to different deployment scenarios. More recently, the OFA² strategy improved the search stage of the OFA framework by taking advantage of the very low cost of sampling already trained sub-networks and by exploring the multi-objective nature of the problem: a set of non-dominated sub-networks are all obtained at once, with distinct trade-offs involving hardware constraints and accuracy. In this work, we propose OFA³, building high-performance ensembles by solving the problem of how to automatically select the optimal subset of the already obtained non-dominated sub-networks. Particularly when components of the ensemble can run in parallel, our results dominate any other configuration of the available sub-networks, taking accuracy and latency as the conflicting objectives. The source code is available at `https://anon-github.automl.cc/r/once-for-all-3-89E3`.

## 1 Introduction

The task of designing neural networks requires a lot of trial and error, as well as a deep understanding of the problem at hand. And even then, there is no guarantee that we will arrive at the optimal architecture. Seeking a way to automate the design of artificial neural architectures, the Neural Architecture Search (NAS) framework emerges [1]. In other words, it includes techniques capable of automatically optimizing the neural network architecture for a given task without human intervention. By automating the process of designing neural networks, we can save a lot of time and resources while potentially arriving at effective architectures. NAS works by searching through a vast space of possible architectures, evaluating their performance, and iteratively refining the search until it finds the best architecture for the given task [1].

To implement the search, NAS frameworks use advanced techniques such as reinforcement learning [2], evolutionary algorithms [3], and gradient-based optimization [4] to properly explore the space of candidate architectures. Additionally, by combining these obtained learning models, it is possible to create a more comprehensive representation that captures multiple levels of abstraction. That is called ensemble learning, which can help compose a diverse set of final architectures that are generated by the NAS process [1]. By combining multiple efficient learning models, the ensemble can explore a wider range of design choices and trade-offs, potentially leading to better overall performance [5]. However, when one considers real-world scenarios, there are often multiple objectives that need to be optimized simultaneously, such as performance, model size, latency, and deployment.

A way to address such a challenge is the Multi-objective Neural Architecture Search (MO-NAS) which considers multiple conflicting objectives simultaneously [6]. MO-NAS typically works by populating a Pareto frontier with distinct trade-offs among multiple objectives, using techniques

such as evolutionary algorithms [3] or progressive search [7]. These candidate architectures are evaluated on a validation set, and the ones that are deemed Pareto-optimal, meaning they cannot be improved on any objective without sacrificing performance on another, are considered efficient solutions and returned to the user.

MO-NAS has the advantage of allowing us to explore a wider range of trade-offs established by conflicting objectives. For instance, we might be able to find architectures that are both highly accurate and efficient [8], or architectures that can perform well on multiple related tasks simultaneously [9]. Also, it can help researchers and practitioners to make more informed a posteriori decisions about the design of their neural networks.

Another factor is the computational cost of generating the optimal architecture. The search process can be computationally expensive and time-consuming, requiring significant resources and infrastructure. Therefore, there is a gain in considering the deployment scenario and available resources as boundary conditions when designing the search process. In OFA² it was presented a multiobjective search with the evolutionary algorithm NSGA-II [10], for both accuracy and latency, consistently outperforming other search strategies, particularly by combining the non-dominated learning models along the Pareto frontier in an ensemble. The work not only finds better architectures in terms of top-1 accuracy and latency but also returns a set of solutions instead of a single one, each of them being optimal considering a specific trade-off among the conflicting objectives. In this work, we propose OFA³, a technique to automatically select the best sub-networks among a population of non-dominated architectures to form even more efficient ensembles. We also provide experiments regarding the latency of the ensemble considering the scenarios of summed and maximum latencies. We show that when the components of the ensemble can run in parallel, thus considering the maximum latency scenario, our results dominate any other configuration of the available single efficient neural networks, resulting in better architectures overall. Notice that the assignment of the acronym OFA³ is motivated by the occurrence of a cascade of three once-for-all mechanisms during the NAS: a single training step, a single search step to populate the Pareto frontier, and a single selection step to compose the ensemble of efficient learning models.

After careful reflection, the authors have determined that this work presents no notable negative impacts to society or the environment.

## 2 Related Works

Due to the diverse composition of our work, we discuss the background by highlighting the three main modules of our approach.

**Neural Architecture Search**. The research field of automating the machine learning process [11], including selecting algorithms, hyperparameters, and architectures, dates back to the early 2000s [12]. However, the focus of those works was mostly on the algorithm and hyperparameter selection, and NAS was not yet a major research topic [1]. In 2019, two new NAS methods were proposed, called Efficient Neural Architecture Search (ENAS) [8] and Once-for-All (OFA) [13]. ENAS involved using a shared weight scheme to reduce the search space, while OFA involved training a large pre-defined network that can be efficiently adapted to different tasks and hardware platforms. ENAS approaches, by sharing weight to reduce the number of parameters, significantly reduce the time and cost required for NAS. While ENAS is computationally efficient, it can be limited by the search space. Meanwhile, OFA is able to find a family of architectures that can perform well across various resource constraints. Their weakness is that it can be challenging to scale to larger datasets or more complex models. Nowadays, NAS has advanced significantly with the development of new search algorithms and optimization techniques. With the ongoings in automating the architecture design process, it is possible to find an expressive number of surveys on NAS [1] [3] [14] [15].

**Multi-objective optimization in NAS.** Emerged as a promising path to address the challenges of traditional NAS methods (e.g., search space, computational cost, generalization, evaluation and interpretability) is the multi-objective optimization approach. Accordingly, it was intuitive to resort to evolutionary multi-objective optimization (EMOO) algorithms as in [10] [16]. Yet, using EMOO algorithms for NAS still lacks an investigation of challenging scenarios such as lack of convexity and presence of many objectives [17]. However, by the late 2010s, multi-objective optimization gained popularity in NAS, and since then, several multiobjective optimization approaches have been proposed, including NSGA-Net [18], DARTS+ [19], and MOEAs [9].

**Evolutionary algorithms.** In the mid to late 2010s, researchers started exploring the use of evolutionary algorithms in NAS. Works like in [20] [21] [22] aimed at adapting the powerful and generic search strategy of meta-heuristics to the NAS specificities. Some methods [23] [24] [25] involved training a controller or a population of architectures using evolutionary algorithms and progressive search, capable of exploring the search space incrementally and sequentially. Other works [26] [27] [28] tried to improve the quality of discrete decisions in the process of searching architectures by utilizing a performance predictor to select promising candidates.

To our knowledge, the existing techniques cannot be scaled to many different deployment scenarios. That means the whole search process must be repeated, thus not being once-for-all, and the model needs to be retrained for different hardware platforms. Also, the overall model size is huge and the footprint is considerable, because the individual trained models do not share weights.

## 3 Methodology

In the traditional Neural Architecture Search (NAS) pipeline, we have three main components: search space, search strategy and performance estimation strategy [29]. We first choose or propose a search space that defines all neural network architectures available to the framework. Then, we choose a search strategy that will guide the exploration and exploitation of the search space. Finally, a performance estimator measures how good the obtained models are and helps to guide the search towards better architectures.

### 3.1 OFA

The Once-for-All (OFA) NAS framework [13] works a little differently than what is found in the traditional NAS pipeline. More specifically, the OFA framework decouples the training and search stages of the NAS pipeline. The biggest advantage of this decoupling is that a single supernetwork is trained only once, and then multiple low-cost searches can be done in this supernetwork finding nested smaller and already trained sub-networks according to different deployment scenarios. The

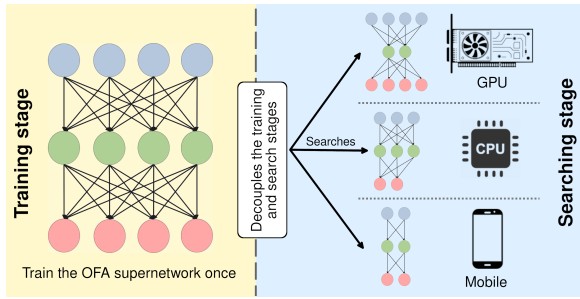

Figure 1: Once-for-All (OFA) framework overview. A single OFA supernetwork is trained only once, and then multiple searches can be performed according to the different deployment scenarios.

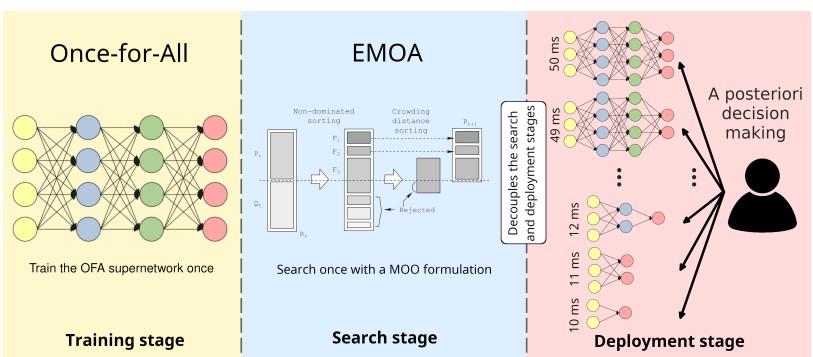

Figure 2: OFA² overview. Instead of manually performing one search for each deployment scenario, an EMOA performs the search of the architectures for all deployment scenarios at once.

deployment scenario can be defined by a hardware constraint, such as latency or FLOPS. Figure 1 illustrates the OFA framework.

### 3.2 OFA²

More recently, a new search strategy called OFA² [30] was proposed for the OFA framework. Instead of performing multiple searches on the OFA supernetwork, the OFA² optimizes the search stage of the OFA framework by solving a multi-objective optimization problem (MOOP) with the use of an evolutionary multi-objective optimization algorithm (EMOA). In other words, a single search is enough to find a representative set of efficient learning models with distinct trade-offs among the conflicting objectives, that being latency and accuracy. Figure 2 illustrates the OFA² framework.

### 3.3 OFA³

Both OFA and OFA² end up with a single or multiple architectures after the search stage of the framework. In this work we aim to give one step further by proposing a strategy to automatically select a subset of the networks found by the OFA² search . This subset of efficient learning models is evolved on a multi-objective perspective to compose a high-performance ensemble. The combinatorial problem of selecting the best subset of efficient learning models is solved here by an EMOA, jointly optimizing the accuracy and latency, by maximizing the former and minimizing the latter. The OFA³ does not focus on the search stage of the framework. Instead, we use the efficient learning models discovered by OFA² during its search stage, and the problem now is how to select among these architectures the best subset that will compose the ensemble.

### 3.4 Ensembles

Ensemble [31] is a technique for combining predictions from multiple individual learning models aiming at a more robust performance. In this work, we propose to solve the problem of determining the number of components in an ensemble and the components themselves as a multi-objective optimization problem, and taking candidates from the Pareto frontier estimated by the OFA² framework. The idea is to find a diverse set of efficient learning models [32], reducing the variance of the error at the output, thus leading to better machine learning models [33]. The strategy of using an evolutionary algorithm to obtain ensembles is not innovative [34]. However, most of the evolutionary approaches for ensembles use the evolutionary process just to create diversity among the individuals of the ensemble and does not propose the ensemble formation problem as a multi-objective and combinatorial problem [35] [36].

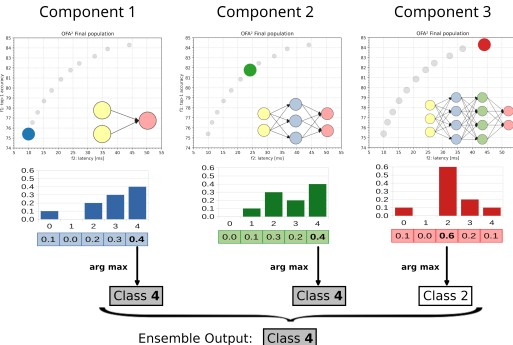

Figure 3: Ensemble output with hard majority voting.

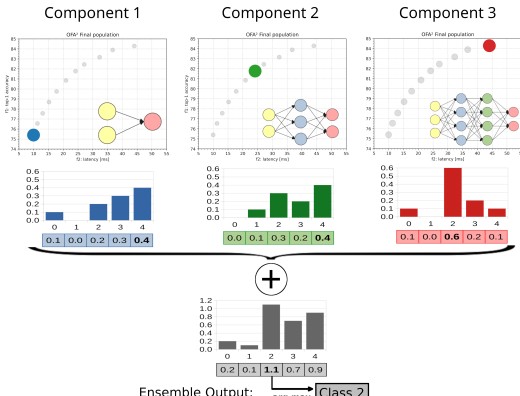

Figure 4: Ensemble output with soft majority voting.

### 3.5 Voting schemes

The simplest way to determine the output of an ensemble is by majority vote. Considering the majority vote ensembles, there are two main approaches for classification problems: the hard voting and the soft voting. In the hard voting, the output of the ensemble is decided according to the most voted top-1 class among the participants of the ensemble. Draw in votes between more than one class must be handled somehow, for example by checking the occurrences of these classes on the second most likely output of each model (top-2 output) and deciding by the most frequent. If there is still a draw in votes, we can keep checking the top-3, top-4, and so on. In this scheme, the output of each neural network has the same importance, regardless how certain the model is about its output. Figure 3 illustrates the hard voting scheme considering an ensemble with 3 components and an image classification problem with 10 classes. The soft voting scheme, on the other hand, takes into account the probability assigned to each class on the output. For this, in case the last layer of the neural networks is not a softmax already (like in the case of the OFA supernetwork), this layer is appended to the last layer of the neural networks. In order to decide the output of the ensemble, we sum the probabilities for each class among all participants of the ensemble, and then take the output with the highest accumulated value. Figure 4 illustrates the soft voting scheme. This helps to alleviate the problem of the hard voting scheme, which is the fact that a vote from a model with a low confidence in its output has the same weight of a vote from a model with a high confidence. This voting scheme provides a way to weight the vote of each architecture according to the confidence of the model in its output class, which can be beneficial in some cases.

### 3.6 Dataset

The dataset used for the training stage of the OFA supernetwork, the search stage of the OFA² strategy, and the ensemble search of the OFA³ is the ImageNet [37], a standard dataset in the computer vision area that consists of 1,281,167 images in the training set, 50,000 images in the validation set and 100,000 images in the test set, organized in 1,000 categories. The training set was used to train the OFA supernetwork. A subset of 10k images of the training set was set apart as a holdout validation set to train the accuracy predictor of the OFA framework, used during the OFA² search. For the OFA³ optimization, a subset with 50k images from the training set was used.

### 3.7 Objective functions

The objective functions optimized are the same as the OFA² search: accuracy and latency. For the accuracy, we use the performance of the ensembles on the validation set of ImageNet, considering the soft voting scheme. For the latency, we use the latency predictor provided by the OFA framework as a latency lookup table [38].

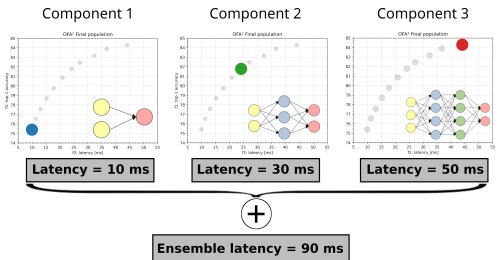

Figure 5: Scenario considering the summed latency.

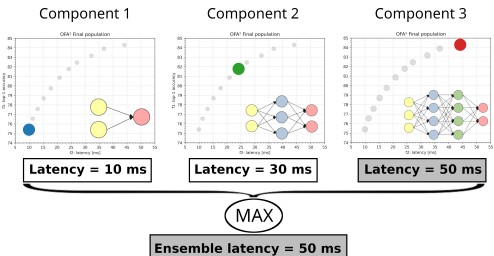

Figure 6: Scenario considering the maximum latency.

### 3.8 Latency

We consider two approaches with respect to the latency of the ensembles: the summed and the maximum latency. In the first approach, the latency of the ensemble is defined to be the sum of all architectures participating in the ensemble. This strategy is based on the premise that we have a single hardware with limited amount of memory to implement the ensemble, and therefore we need to load each model one at a time to evaluate its performance. Figure 5 illustrates this scenario considering an ensemble with 3 neural networks. In the second approach, the latency of the ensemble is equal to the model's latency that has the highest value among those networks participating in the ensemble. This strategy is based on the premise that parallelization is viable, and therefore all models can be evaluated simultaneously. Figure 6 illustrates this scenario considering the same ensemble with 3 architectures.

### 3.9 Candidate architectures

The neural networks candidates to participate on the ensembles are the 100 efficient learning models produced by the OFA² search. Figure 7 shows the performance of these efficient architectures (in red) in the objective space of the multi-objective formulation, properly characterizing a Pareto frontier.

## 4 Experiments

The implementation of the code was done with the pymoo multi-objective optimization framework [39] and the PyTorch framework [40]. For the evaluation of the neural network architectures, an NVIDIA Quadro RTX 8000 with 48 GB of memory was used. We consider both the sum and maximum latency approaches, but only the soft voting scheme was used to decide the output of the ensemble.

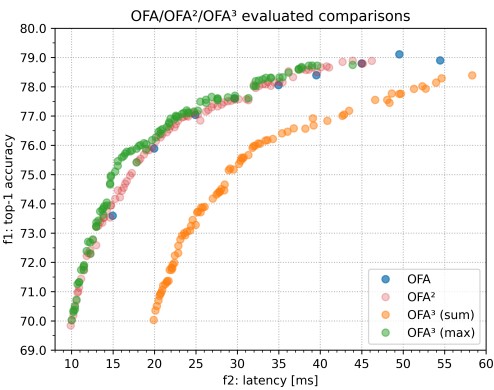

Figure 7: OFA, OFA² and OFA³ architectures.

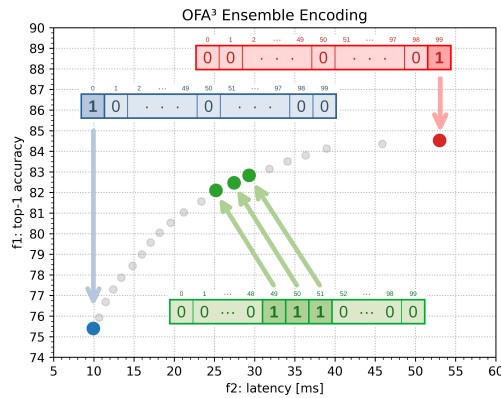

Figure 8: Encoding used to search the ensembles.

### 4.1 Encoding

The encoding used is a simple array with 100 binary genes, one for each neural network from the pool of 100 candidate architectures to form the ensemble, meaning that if a specific gene has a value 0, then the neural network represented by that gene does not compose the ensemble, and if the gene has a value of 1, then the corresponding learning model is part of the ensemble. This gives us a search space of $2^{100}$ combinations. Figure 8 illustrates three examples of encodings and their respective sets of neural networks participating in the ensembles. The encoding in blue has a single gene with the value 1 in its first position, meaning that only the first neural network of the population will compose the ensemble. The encoding in green has three consecutive genes with the value of 1 in the middle of the array, meaning that these three models represented by these genes will take part in the ensemble. The encoding in red has all genes at value 0, except the last one, meaning that only the last neural network of the population will be included in the ensemble.

### 4.2 The solver

We adopted three solvers for the optimization: the NSGA-II [10], the SMS-EMOA [41] and the SPEA2 [42], although others EMOA algorithms could also be used. Four operators are used during the iterations of evolutionary algorithms: sampling, mutation, crossover and selection. The sampling operator is related to the initialization of the algorithm, and we decided to start it with all individuals of the population having all their 100 genes set to one. This means that all candidate neural networks compose the ensemble at the initial population, being the most inefficient scenario but revealing the role performed by each candidate learning model. The mutation operator used is the bitflip with probability of 1%. The crossover operator (also known as recombination) used is the uniform crossover, which means that the value of each gene of the child solution is randomly taken from one of the parents' solution with equal probability. Finally, the selection operator defines a criterion for choosing the individuals of the current population that will be used to generate the offspring, that is, the next generation of individuals. The algorithms ran for a total of 1,000 generations for the summed latency and 2,000 generations for the maximum latency. Three different random seeds were used for each algorithm. The results are discussed in the next section.

## 5 Results

Table 1 show the results for the OFA, OFA² and OFA³ searches. We can see that for most of the latency constraints, the OFA³ considering the maximum latency performs better than OFA and OFA². For the highest latencies, the OFA and OFA² perform better than OFA³ though. This could be explained by the fact that the evolutionary algorithm populates mid and lower latencies regions with more individuals than higher latencies, which could be alleviated by performing a local search in this region. These results can also be seen in Figure 7. The computational costs of all methods are negligible when compared with the cost of training the Once-for-All supernetwork (1,200 GPU hours). Next, we show the progression of the populations in details for both the summed and the maximum latencies scenarios.

Table 1: Comparison of ImageNet results between different hardware-aware NAS search methods.

| method | \multicolumn{9}{c}{ImageNet Top-1 % under latency constraints} | Search cost |
| | 10 ms | 15 ms | 20 ms | 25 ms | 30 ms | 35 ms | 40 ms | 45 ms | 50 ms | GPU hour |
| --- | --- | --- | --- | --- | --- | --- | --- | --- | --- | --- |
| OFA | N/A | 73.60 | 75.89 | 77.04 | 77.61 | 78.06 | 78.39 | 78.80 | **79.11** | 0.83 |
| OFA² | 69.84 | 73.95 | 75.94 | 77.11 | 77.56 | 78.17 | 78.66 | **78.89** | 78.89 | **0.01** |
| OFA³ (sum) | N/A | N/A | 70.03 | 73.59 | 75.21 | 76.22 | 76.68 | 76.34 | 77.76 | 0.98 |
| OFA³ (max) | **70.03** | **74.97** | **76.18** | **77.20** | **77.70** | **78.31** | **78.73** | 78.75 | 78.75 | 1.83 |

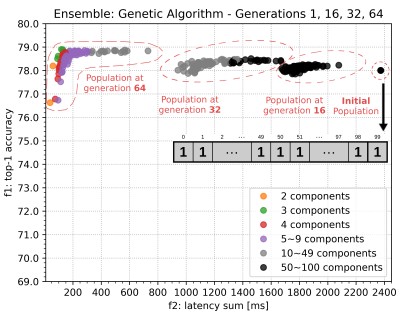

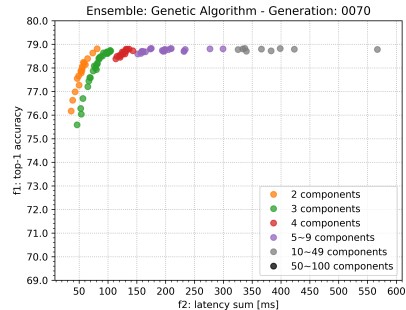

(a) Progression of the initial populations.

(b) Population at generation 70.

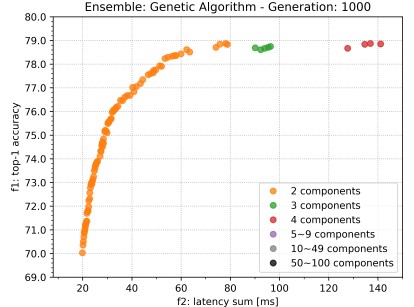

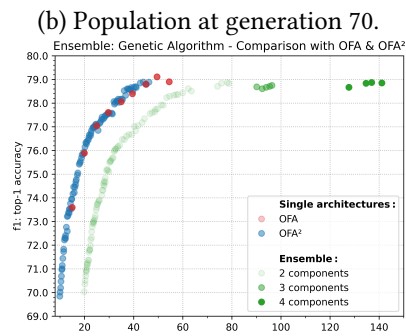

(c) Final population at generation 1,000.

(d) OFA, OFA² and OFA³ results comparison.

Figure 9: Progression of the NSGA-II populations of individuals reprensenting ensembles for the summed latencies approach. (a): Generations 0, 16, 32 and 64. (b): Generation 70. (c): Last population at generation 1,000 (d): Comparison with OFA and OFA² architectures.

## 5.1 Summed latency

For the summed latency scenario, the only restriction used during the optimization is that an individual must have at least two genes with value one, meaning that no single architectures are allowed. The single black point on the extreme right of Figure 9a illustrates the initial population. We have just a single point because all ensembles represented by this first population are equal (since the initialization starts with all genes of all ensembles equal to one), having therefore the same accuracy and latency. In the same figure, we can see the populations for generations 16, 32 and 64, represented by the different clouds of points, as indicated. These generations were chosen as power of two to illustrate the non-linear characteristic of the evolution. At generation 16, only ensembles with 50 or more components are present. At generation 32, most of the ensembles have between 10 and 49 components, and at generation 64, we start seeing ensembles with 2, 3 and 4 neural networks. We then plot the individuals at generation 70 in Figure 9b (please note the change of scale in the latency axis).

In Figure 9c we can see the final population after 1,000 iterations. Finally, Figure 9d illustrates a comparison between the results of the final population of ensembles using the summed latency against the single architectures obtained from OFA and OFA² searches, the latter one being used as the foundation of the ensembles. We can see that the ensembles found approximates the Pareto-front for a multi-objective optimization problem with two conflicting objectives. All of these ensembles are, however, dominated by the single architectures found by OFA² or OFA searches. This dominance of single architectures can be explained due to the difference of scale between the objective functions. Take for example the two smallest neural networks as components of the ensemble. The first one presents a latency of 9.9 ms and accuracy of 69.84 %, while the second one presents a latency of 10.0 ms and accuracy of 70.02 %. When summing the latencies of these two architectures, we have a total latency of almost 20.0 ms. If we take the individual architecture with

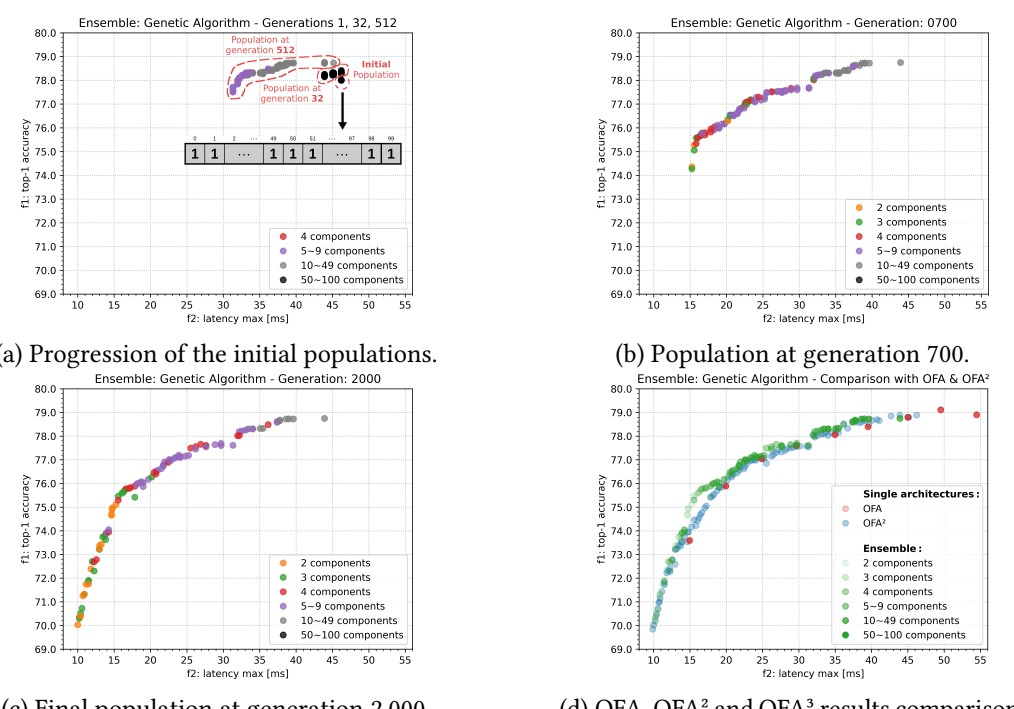

(a) Progression of the initial populations.

(b) Population at generation 700.

(c) Final population at generation 2,000.

(d) OFA, OFA² and OFA³ results comparison.

Figure 10: Progression of the NSGA-II populations of individuals reprensenting ensembles for the maximum of latencies approach. (a): Generations 1, 32 and 512. (b): Generation 700. (c): Last population at generation 2,000. (d): Comparison with OFA and OFA² architectures.

the highest latency under 20 ms, we have a single neural network with 75.94 % of accuracy and 19.7 ms of latency. It is unlikely that and ensemble of two architectures with around 70 % of accuracy surpass the 76 % accuracy of the single architecture with equivalent latency, even more considering that these two neural networks of around 70 % of accuracy are probably similar to each other on the decision space. We argue that the sum of latencies penalize too much the ensembles, and that in this scenario, it may be better to use single architectures instead.

## 5.2 Maximum latency

To alleviate the problem of difference in scale between latency and accuracy presented during the summed latency approach, we propose the same experiment but taking the maximum latency of the neural networks participating on the ensemble to be the latency of the ensemble itself. The lowest black point in Figure 10a illustrates the initial population, again with all genes equal to 1, meaning that all candidate neural networks compose the initial ensembles. In the same figure we see two clouds of points illustrating the populations at generations 32 and 512, as indicated. At generation 32 we still have only ensembles with more than 50 components, while at generation 512 the ensembles present less than 50 neural networks. Figure 10b illustrates the population at generation 700 showing ensembles with different number of components. Figure 10c shows the last population of ensembles, after 2,000 generations. On the contrary to what was done with the summed latency approach, here we do not restrict the optimization to ensembles with 2 components or more. In fact, we can see that in the final population there are some individuals that are single architectures (in orange), meaning that other ensembles with more neural networks with lower latency perform actually worse than that specific single neural network.

Figure 10d compares the ensembles found by the evolutionary algorithm against the OFA and OFA² single architectures. Here we can clearly see advantages of the ensembles over the single

architectures, with the former dominating the latter for almost all the latency range. The only exception happens at the beginning of the curve, where the accuracies and latencies of ensembles and single architectures are similar, and at the very end of the curve, where single architectures dominates the ensembles. In fact, some of the ensembles found by the evolutionary algorithm in the region of lower latencies are actually single architectures. This can be explained since the pool of neural networks to form the ensembles increases proportionally with the latency of the ensembles. For example, for the ensemble with the highest latency, all neural networks are available to join the ensemble, while at the ensemble with the lowest latency, there is no ensemble at all, with only one neural network being available to form the "ensemble". The degraded performance of the OFA³ selection method on the region of higher latencies can be explained by the fact that the evolutionary algorithm populates the region with intermediary latencies with more individuals than other regions. This could be alleviated by performing a local search on these specific regions, or by imposing some restriction during the optimization.

It is interesting to note that the evolutionary algorithm tends to reduce the number of neural networks of the ensembles with the generations, even though the first generation started with all neural networks being part of the ensembles. This indicates that ensembles with fewer components may perform better than ensembles with all neural networks [43]. The computational burden of the OFA³ search and of starting with full ensembles is not relevant, given that we have to run each efficient learning model produced by OFA² only once to allow any kind of voting configuration.

# 6 Concluding Remarks

In this work, we presented OFA³, an extension of OFA² [44]. The starting principle was provided by OFA [13], which has promoted the decoupling of training and search stages in NAS, thus making the search stage of negligible cost, when compared to the Once-for-All training of the super-network. In fact, any sub-network that is sampled from the search space is already trained, thus making of low cost even a more elaborate search procedure. Therefore, there is room for the multi-objective search performed by OFA² and, in this paper, for the additional multi-objective selection performed by OFA³. The OFA³ proposal involves a cascade of three once-for-all mechanisms during the NAS: a single training step (provided by OFA), a single search step to populate an approximation of the Pareto frontier (provided by OFA²), and a single selection step over the output of OFA² to compose the ensemble of efficient learning models. This is a remarkable achievement due to two main reasons: (1) the whole computational cost for the search stage remains of a reduced amount when compared to the once-for-all training of the super-network, even performing a cascade of two consecutive multi-objective searches; (2) The multi-objective selection of efficient components (taken from the output of OFA²) for the ensemble, which is the main contribution of OFA³, is motivated by three main factors: (2.1) the guaranteed presence of distinct trade-offs among the candidate components provided by OFA², given that they populate an approximation of the Pareto frontier; (2.2) the assurance that they are independent models and can operate fully in parallel; (2.3) the possibility of automatically choosing just a subset of the efficient models produced by OFA² as components of the best ensemble. Those are the main motivation to support the gain in performance when compared with, for instance, a single model of the same size of the whole ensemble, which would not be implementable by resorting to independent fully parallelizable sub-networks. This framework discovers architectures with improved top-1 accuracy and latency. All the source code has been made available and we show in the experiments that OFA³ compares favorably with the architectures found by the original OFA and OFA², in the sense of achieving higher accuracy for the same latency threshold, supposing that the components of the ensemble are run in parallel, given that they are independent models. Additionally, the evolutionary algorithm adopted by OFA³ is able to determine the appropriate number of components of the ensemble, being a subset of the efficient models provided by OFA², while keeping constraints (such as latency) within specific bounds along the Pareto frontier.

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
