# OpenReview forum: "OFA³: Automatic selection of the best non-dominated sub-networks for ensembles"
_automl.cc/AutoML/2023/Conference — Submitted to AutoML 2023_

### Official Review · Reviewer_TFUH · 2023-04-08

**Potential Impact On The Field Of Automl Rating:** 2
**Technical Quality And Correctness Rating:** 2
**Clarity Rating:** 2

**Summary Of Contributions:**

The authors have introduced OFA3, which is an extension of the OFA and OFA2 NAS frameworks. OFA3 utilizes a multi-objective optimization approach to select neural networks for forming ensembles, and it can automatically identify efficient ensembles from OFA2. This approach helps in discovering architectures that exhibit better top-1 accuracy and latency. The authors have also conducted experiments related to ensembles, including tests with encodings and operators in the EA. According to the experimental results, OFA3 outperforms OFA2 and OFA in most cases while also reducing computational costs.

**Actions Required To Increase Overall Recommendation:**

- Fill in a comparison table for the search time.
- Add experiments comparing with OFA and OFA2.
- Compare with other multi-objective methods.

**Clarity:**

It seems that the authors should make their contributions more clear in the Abstract and Introduction sections. It is difficult to understand the purpose for which they are presenting this methodology after reading these two sections. Also it would also be helpful to reduce the number of abbreviations, as there seem to be too many of them.

**Overall Review:**

Positive aspects: The paper attempts to address the ensemble formation problem as a multi-objective problem, which is a novel approach. The internal ablation study that compares sum latency, max latency, and voting scheme is also a valuable contribution.

Negative aspects: Upon reading the paper, it is difficult to understand why this methodology is necessary. Additionally, the explanation of the multi-objective problem is excessive and should be condensed into a single section. The formulation of the methodology is lacking in detail and could benefit from a more thorough and reader-friendly (or mathematical) presentation. Lastly, the paper lacks experimental results to support the claims made in the conclusion. It would be beneficial to provide metrics such as computational cost or performance improvement in a table format.

**Potential Impact On The Field Of Automl:**

While using ensembles in MONAS is a promising attempt, it does not seem to be an innovative approach. Additional evidence is needed to support why this approach is considered good.

**Review Confidence:**

3: You are fairly confident in your assessment. It is possible that you did not understand some parts of the submission or that you are unfamiliar with some pieces of related work.

**Review Rating:**

5: Borderline Leaning Reject: Technically sound paper where reasons to reject nonetheless outweigh reasons to accept. Please use sparingly.

**Review Summary:**

The evidence presented for why the proposed methodology is necessary may not be sufficient to convince readers. To better support the claims, the authors should provide additional results such as direct comparisons with OFA and OFA2, the extent of performance improvement, and the computational cost aspect mentioned in conclusion. This would help in providing a more convincing argument for the necessity of the proposed methodology.

**Technical Quality And Correctness:**

In the conclusion, the authors claim that the OFA3 searched architecture consistently performs better than the architectures found by the original OFA and the OFA2 search in most cases while being computationally cheaper. However, there is a lack of direct comparison with the two baselines, and there is also a shortage of comparisons from the perspectives of search efficiency and computational cost.
It would be beneficial to not only explain why the proposed method outperforms the two baselines but also to provide experimental evidence to support the claim. Additionally, the authors should provide a more detailed and reader-friendly introduction to OFA2.

---

> ### Author Response · Authors · 2023-05-06
> **Response to Reviewer TFUH**
>
> Dear reviewer TFUH, thank you for your time and your valuable comments addressed to our work. We adjusted our paper taking your comments into consideration. Next, we copied some of your comments and added our reply straight after it:
>
> - [While using ensembles in MONAS is a promising attempt, it does not seem to be an innovative approach. Additional evidence is needed to support why this approach is considered good.]
> - [… it is difficult to understand why this methodology is necessary.]
>
> The OFA³ proposal involves a cascade of three once-for-all mechanisms during the NAS: a single training step (provided by OFA), a single search step to populate the Pareto frontier (provided by OFA²), and a single selection step to compose the ensemble of efficient learning models, provided by OFA³. This is a remarkable achievement due to two main reasons: (1) OFA has split training and searching the neural architecture, thus making the search stage of negligible cost, when compared to the training of the super-network. Therefore, there is room for more elaborate search strategies, as performed by OFA² and OFA³, both being of a multi-objective nature and cascading. (2) Making a multi-objective selection of efficient components for the ensemble (being the main contribution of OFA³) is motivated by three main factors: (2.1) the guaranteed presence of distinct trade-offs among the candidate components provided by OFA², given that they populate an approximation of the Pareto frontier; (2.2) the assurance that they are independent models and can operate fully in parallel; (2.3) the possibility of automatically choosing just a subset of the efficient models produced by OFA² as components of the best ensemble. Those are the main motivation to support the gain in performance when compared with, for instance, a single model of the same size of the whole ensemble, which, to make matters worse, would not be implementable by resorting to independent fully parallelizable sub-networks. We have included this additional discussion at the concluding remarks of the paper.
>
> - [… there is also a shortage of comparisons from the perspectives of search efficiency and computational cost.]
> - [Lastly, the paper lacks experimental results to support the claims made in the conclusion. It would be beneficial to provide metrics such as computational cost or performance improvement in a table format.]
>
> We have included additional information regarding computational cost, thus allowing a direct comparison. Notice that the search stage of OFA is of very low computational cost, opening the possibility of investing resources to promote more elaborate and multi-objective search stages for OFA² and OFA³.
>
> - [Additionally, the authors should provide a more detailed and reader-friendly introduction to OFA².]
>
> Given that we have one additional page to prepare the final version of the paper, we have enriched the presentation of OFA², at Section 1 (Introduction) and at Section 3 (Methodology).
>
> - [Compare with other multi-objective methods.]
>
> We extended our work and included other evolutionary multi-objective optimization algorithms (EMOA) to perform the search. We now run our experiments with the NSGA-II, SMS-EMOA and SPEA2 algorithms. For each of these algorithms, we ran the search for 3 different random seeds. We also provided Jupyter notebooks (with outputs) for these comparisons:
>   1. notebook to run OFA² search: [ofa2.ipynb](https://anon-github.automl.cc/r/once-for-all-3-89E3/jupyter-notebooks/ofa2.ipynb)
>   2. notebook to run OFA³ search (summed latency): [ofa3-summed-latency-N1000.ipynb](https://anon-github.automl.cc/r/once-for-all-3-89E3/jupyter-notebooks/ofa3-summed-latency-N1000.ipynb)
>   3. notebook to run OFA³ search (maximum latency): [ofa3-max-latency.ipynb](https://anon-github.automl.cc/r/once-for-all-3-89E3/jupyter-notebooks/ofa3-max-latency.ipynb)
>   4. notebook to plot figures and results of OFA³: [ofa3-plot.ipynb](https://anon-github.automl.cc/r/once-for-all-3-89E3/jupyter-notebooks/ofa3-plot.ipynb)
>
> Additionaly, the OFA performance has already been contrasted, in its original proposal, with the performance of a high number of relevant contenders, with a clear advantage both in terms of accuracy and computational burden. So, making a direct comparison solely with OFA and OFA² is a valid methodology here, because OFA was already properly positioned before relevant contenders in the literature.

---

### Official Review · Reviewer_6Qzm · 2023-04-11

**Potential Impact On The Field Of Automl Rating:** 2
**Technical Quality And Correctness Rating:** 2
**Clarity Rating:** 3
**Actions Required To Increase Overall Recommendation:** (already addressed above)

**Summary Of Contributions:**

This paper proposes the framework OFA^3 to select efficient ensembles out of an already obtained set of single neural networks. It optimizes for accuracy and latency by utilizing a well-established evolutionary multi-objective optimization algorithm.

**Clarity:**

The paper is well structured. It clearly elaborates on the components used in OFA^3. However, it would be of interest to shift the focus from extensively discussing familiar concepts, such as voting schemes or encodings, to providing deeper insights into the framework's behavior during the search process.

Also, for plotting the performance of the ensembles during the search process, choosing generations 14, 32, 60, and 66 for sum latency and 38, 179, and 212 for maximum latency seems arbitrary (given that there are 1000 generations in total) and has not been further elaborated on. Other inconsistencies, such as Figure 9 (a) stating that the results are based on "hard voting" in contrast to Line 230 stating that only the soft voting scheme has been considered, should be avoided as well.

**Ethics Details (Optional):**

No potential societal impacts of the work have been discussed.

**Overall Review:**

The paper gives a clear introduction into the topic and a good overview of related work. The methodology primarily explores established concepts, which, while informative, offers limited originality. The results are insightful but suggest that the performance of the final ensembles is reported on the same dataset that they were searched on, thus could be overfitting.

**Potential Impact On The Field Of Automl:**

OFA^3 relies on existing neural architectures (for their experiments, 100 architectures have been utilized that were obtained by the existing search strategy OFA^2). Also, it makes use of well-established soft voting schemes to determine the output of the ensembles, and takes the same objective function as proposed for OFA^2. Thus, the primary novelty of this paper is the formulation of the ensemble formation as a multi-objective problem. The authors have chosen the existing multi-objective genetic algorithm NASGA-II for this problem.

Although the resulting ensembles achieve a higher top-1 accuracy than the single neural architectures from which they are composed of, the performance improvement is modest. It could be observed that the performance only improves in case of optimizing for the single maximum latency of all ensemble members instead of summing the latency of all ensemble members. Additionally, the paper indicates that the performance results have been reported for the dataset on which the ensembles were searched. So, it cannot be ruled out that the ensembles overfit on this validation set for which they were optimized.

Given these facts, the overall potential impact on the filed of AutoML is considered as small.

**Review Confidence:**

4: You are confident in your assessment, but not absolutely certain. It is unlikely, but not impossible, that you did not understand some parts of the submission or that you are unfamiliar with some pieces of related work.

**Review Rating:**

3: Reject: For instance, a paper with technical flaws, weak impact, and/or weak evaluation.

**Review Summary:**

The primary contribution of this paper is the formulation of a multi-objective problem to form ensembles with pre-existing neural networks. The resulting performance improvements are modest and the overall impact of the framework is limited.

**Technical Quality And Correctness:**

The proposed approach solely makes use of already existing methods and now flaws have been perceived in doing so. However, the performance of the ensembles should be evaluated on another held-out dataset for which they have not been searched for, since the ensembles might overfit on the validation set for which they were searched for.

---

> ### Author Response · Authors · 2023-05-06
> **Response to Reviewer 6Qzm**
>
> Dear reviewer 6Qzm, thank you for your time and your valuable comments addressed to our work. We adjusted our paper taking your comments into consideration. Next, we copied some of your comments and added our reply straight after it:
>
> - [Additionally, the paper indicates that the performance results have been reported for the dataset on which the ensembles were searched. So, it cannot be ruled out that the ensembles overfit on this validation set for which they were optimized.]
>
> In fact, the text was inducing the reader to a misinterpretation of the way in which the dataset was split. We have rechecked the experiments and fortunately the issue was restricted to the unclear description at the body of the paper. We are now providing the correct explanation: all ensembles are being evaluated on another held-out dataset for which they have not been searched for. The search was done on a subset of the training set with 50k images. The evaluation was done on the 50k images of the validation set that were not used whatsoever during the search.
>
> - [… the overall potential impact on the field of AutoML is considered as small.]
>
> The OFA³ proposal involves a cascade of three once-for-all mechanisms during the NAS: a single training step (provided by OFA), a single search step to populate the Pareto frontier (provided by OFA²), and a single selection step to compose the ensemble of efficient learning models, provided by OFA³. This is a remarkable achievement due to two main reasons: (1) OFA has split training and searching the neural architecture, thus making the search stage of negligible cost, when compared to the training of the super-network. Therefore, there is room for more elaborate search strategies, as performed by OFA² and OFA³, both being of a multi-objective nature and cascading. (2) Making a multi-objective selection of efficient components for the ensemble (being the main contribution of OFA³) is motivated by three main factors: (2.1) the guaranteed presence of distinct trade-offs among the candidate components provided by OFA², given that they populate an approximation of the Pareto frontier; (2.2) the assurance that they are independent models and can operate fully in parallel; (2.3) the possibility of automatically choosing just a subset of the efficient models produced by OFA² as components of the best ensemble. Those are the main motivation to support the gain in performance when compared with, for instance, a single model of the same size of the whole ensemble, which, to make matters worse, would not be implementable by resorting to independent fully parallelizable sub-networks. We have included this additional discussion at the concluding remarks of the paper.
>
> - [Also, for plotting the performance of the ensembles during the search process, choosing generations 14, 32, 60, and 66 for sum latency and 38, 179, and 212 for maximum latency seems arbitrary (given that there are 1000 generations in total) and has not been further elaborated on.]
>
> Those portraits along the evolutionary process have been chosen aiming at illustrating the behavior of the evolutionary process along the search. We now changed most of the generations to power of 2 values, so we can explicitly illustrate the non-linear characteristic of the evolution. The generations for the summed latency were replaced from {14, 32, 60, 66} to {16, 32, 64, 70}, while the generations for the maximum latency were replaced from {38, 179, 212} to {32, 512, 700}. We have followed the evolutionary behavior and made the decision of which portrait to get based on a visual comparison of scenarios. The arbitrary nature of the choice is now mentioned in the text.
>
> - [Other inconsistencies, such as Figure 9 (a) stating that the results are based on "hard voting" in contrast to Line 230 stating that only the soft voting scheme has been considered, should be avoided as well.]
>
> This and other inconsistencies have been fixed after a careful proofreading.

---

> > ### Comment · Reviewer_6Qzm · 2023-05-08
> > **Response to Authors**
> >
> > Thank you for your response.
> > Concerning the most crucial point (3):  it is important to note that the paper presents OFA^3, but does not present the cascade of three once-for-all mechanisms. It is explicitly stated that OFA as well as OFA^2 have been proposed for the OFA framework in prior work (lines 121 and 129). Consequently, the primary contribution of this paper is confined to the selection step of composing ensembles, which essentially revolves around formulating the multi-objective problem (accounting for accuracy and latency).
> > In light of this persisting concerns, I will maintain my current score.

---

> > > ### Author Response · Authors · 2023-05-09
> > > **Response to Reviewer 6Qzm**
> > >
> > > Dear reviewer 6Qzm, thank you again for your response. The selection step for optimally composing ensembles of efficient learning models, the main contribution of OFA3, is motivated by the previous existence of diverse trade-offs provided by the outcome of OFA2. It is not an incremental contribution, given that it: (1) involves a multi-objective search that is capable of automatically determining the number of components (**innovation**); (2) requires an additional computational cost that is only viable due to the extremely low cost involved when manipulating OFA efficient learning models (**opportunity**); (3) guides to state-of-the-art performance in the objective space (**competitiveness**); (4) adds robustness to the performance metrics (variance reduction), a well-known contribution of committee machines (**theoretical guarantee**). We are strictly following the page limit for the new version of the paper. The allowed extra page was devoted to reacting to the reviewers' feedback, according to the "Formatting Instructions" section of the [Call for Papers](https://2023.automl.cc/calls/forpapers). We have: (1) added a table of results with a comparison to the baselines; (2) increased the size of the figures; (3) enriched Section 1 (Introduction) and Section 3 (Methodology); (4) provided a better discussion in Section 6 (Concluding Remarks). All these editing initiatives were to improve the paper presentation, following the feedback given by the reviewers.

---

### Review · Reproducibility_Reviewer_PAWL · 2023-04-12

**Completeness Of Code And Dataset Supplement Rating:** 1
**Usability And Ease Of Reproducibility Rating:** 1
**Actions Required To Increase The Reproducibility And Overall Recommendation:** See Review Summary

**Completeness Of Code And Dataset Supplement:**

No code is provided.

**Overall Reproducibility Review:**

The paper is completely irreproducible without code.

**Review Confidence:**

4: You are confident in your assessment, but not absolutely certain. It is unlikely, but not impossible, that you did not understand some parts of the submission or that you are unfamiliar with some pieces of the code or data.

**Review Rating:**

1: Very Strong reject, no code has been made available.

**Review Summary:**

The reproducibility review is a very strong reject until code is provided and the checklist is answered properly.

**Summary Of Necessary Code And Dataset Supplement:**

The paper presents OFA3, an extension of the OFA and OFA2 NAS frameworks that formulates the selection of neural networks to form ensembles as a multi-objective optimization problem.
The authors use an evolutionary multi-objective optimization algorithm (EMOA) to jointly optimize accuracy and latency. The NSGA-II algorithm is used for this purpose.
The OFA^2 algorithm from Ita et al., 2023 is used for finding the 100 candidates for the ensemble. Single subnets found using the OFA^2 algorithm are also used as baselines.
The pymoo multi-objective optimization framework is used to perform the searching.
The ImageNet dataset is used for all experiments.

**Usability And Ease Of Reproducibility:**

No code is provided.

---

### Official Review · Reviewer_z1tG · 2023-04-21

**Potential Impact On The Field Of Automl Rating:** 2
**Technical Quality And Correctness Rating:** 2
**Clarity Rating:** 3

**Summary Of Contributions:**

The authors propose an ensemble technique, OFA3, which addresses the problem of automatically selecting the optimal subset of the already obtained non-dominated subnetworks.
OFA3 is an extension of the OFA and OFA2-NAS frameworks that formulates the selection of neural networks to form ensembles as a multi-objective optimisation problem and automatically finds efficient ensembles from OFA2.

**Actions Required To Increase Overall Recommendation:**

- Please improve the presentation of the figures. These are minor issues, but their increased occurrence deteriorates the good impression.
- Fix the naming issues of chapter 4 and 5.
- The application needs an overhaul. In particular, a comparison over several runs and/or initialisations.


**Clarity:**

In most places the text is legible and clear, but there are sections where clarity is affected.
- A point of criticism is the structure or coherent presentation of the content. For example, chapter 4 talks about "Experiments & Results" but only presents the algotithm and "other details" and is followed by another chapter on "Results".
- Figures 1, 2, 3, 4, 5, 6, 9 and 10 are difficult or impossible to read due to their presentation, in particular the size chosen. Even if this is due to format requirements, the illustrations should be adjusted and improved.
- A small point regarding Figures 9 and 10: please keep the axis labels constant (please do not change decimal and integer)
- The size of the graphs should be constant (the middle of the three graphs is larger than the outer two).
- Similarly, in Figure 7, please keep the axis scales constant (x and y axis).

**Overall Review:**

The paper represents an interesting evolution of existing methods.
The performance and results presented show potential for future application.
However, the overall impression is marred by occasional formal errors.
Furthermore, the application raises questions that need to be clarified in order to allow a better assessment of the method and its performance.


**Potential Impact On The Field Of Automl:**

While the premise of the paper and the aim are very interesting, questions arise about the impact on the scientific community.
The paper presents itself as a worthwhile evolution of the OFA and OFA2 approach, and can provide some evidence for this in the text.
In general, the method is more of a refinement of already established methods.
Although it suggests that it is better than other methods on the core issues of computational cost, it lacks hard facts such as computation time, etc. This, combined with other shortcomings, reduces the overall impression and thus the likelihood of widespread acceptance and use by scientists and practitioners alike.


**Review Confidence:**

4: You are confident in your assessment, but not absolutely certain. It is unlikely, but not impossible, that you did not understand some parts of the submission or that you are unfamiliar with some pieces of related work.

**Review Rating:**

3: Reject: For instance, a paper with technical flaws, weak impact, and/or weak evaluation.

**Review Summary:**

Unfortunately, the positive impressions of the paper are marred by the formal errors and criticisms of the application. In particular, small details in the formal presentation, such as errors in chapter titles or careless mistakes in the graphics, detract from the good impression. In addition, the sometimes superficial presentation in the experiments and method section detract from the overall impression.

**Technical Quality And Correctness:**

The main problems with the paper come from the methods and application sections.

Methods:
- In the methods section, the method itself is only superficially presented and would be improved by a more explicit presentation of the processes and theoretical underpinnings.
- Another point of criticism is the structure or coherent presentation of the content. For example, chapter 4 talks about "Experiments & Results" but only presents the algotithm and "other details" and is followed by another chapter on "Results".

Another point of criticism concerns the application or example.
- The authors themselves say that the sampling operator used is related to the initialisation of the algorithm. Here it would be important to show how different initialisations affect the sampling and thus the result.
- Furthermore, a fixed initialisation with all "genes set to one" is given, for which a conclusive justification is also given. However, the question arises what happens if this initialisation is NOT given, does the result change and if so, how much?
- A final point here is that when doing experiments that are stochastic in nature, please do several runs and give their average results and associated standard deviations rather than just a single run.

General remarks:
- In addition, the paper lacks a self-critical assessment and presentation of possible disadvantages of the method.
- The authors claim that their method is better in terms of computational resources. This argument is also understandable however no clear evidence is given in terms of recorded computation time or respective comparisons.

---

> ### Author Response · Authors · 2023-05-06
> **Response to Reviewer z1tG**
>
> Dear reviewer z1tG, thank you for your time and your valuable comments addressed to our work. We adjusted our paper taking your comments into consideration. Next, we copied some of your comments and added our reply straight after it:
>
> - [Although it suggests that it is better than other methods on the core issues of computational cost, it lacks hard facts such as computation time, etc.]
> - [The authors claim that their method is better in terms of computational resources. This argument is also understandable however no clear evidence is given in terms of recorded computation time or respective comparisons.]
>
> We have included additional information regarding computational cost, thus allowing a direct comparison in Table 1. Notice that the search stage of OFA is of very low computational cost, opening the possibility of investing resources to promote more elaborate and multi-objective search stages for OFA² and OFA³.
>
> - [A final point here is that when doing experiments that are stochastic in nature, please do several runs and give their average results and associated standard deviations rather than just a single run.]
>
> The experiments were redesigned to consider the average of multiple runs. We now considered three different evolutionary algorithms for the search: NSGA-II, SMS-EMOA and SPEA2. Each of these algorithms ran for 3 times with 3 different random seeds (the Jupyter notebook with these runs is available [here](https://anon-github.automl.cc/r/once-for-all-3-89E3/jupyter-notebooks/ofa3-max-latency.ipynb) for the maximum latency approach, and [here](https://anon-github.automl.cc/r/once-for-all-3-89E3/jupyter-notebooks/ofa3-summed-latency-N1000.ipynb) for the summed latency approach).
>
> - [… the paper lacks a self-critical assessment and presentation of possible disadvantages of the method.]
>
> We have improved the paper so that the possible disadvantages of the proposed OFA³ methodology were left clear: (1) Additional computational burden to perform the two multi-objective searches; (2) Necessity of an additional partition of the training dataset to perform the selection of ensemble components; (3) The OFA³ search may be beneficial only for the maximum latency scenario. For the summed latency scenario, it may be better to use single architectures instead of ensembles.
>
> - [Please improve the presentation of the figures. These are minor issues, but their increased occurrence deteriorates the good impression.]
> - [Fix the naming issues of chapter 4 and 5.]
> - [The application needs an overhaul. In particular, a comparison over several runs and/or initialisations.]
>
> All these three final requirements have been fully incorporated into this new version of the paper.

---

### Official Review · Reviewer_tDcb · 2023-04-21

**Potential Impact On The Field Of Automl Rating:** 2
**Technical Quality And Correctness Rating:** 2
**Clarity Rating:** 3

**Summary Of Contributions:**

This paper proposes OFA³, a strategy to build high-performance ensembles for multi-objective architecture search. This paper is based on OFA², which uses the evolutionary search algorithm NSGA-II for accuracy and latency multi-objective search and returns a set of solutions. The proposed method further improves on the latter search method by selecting the best sub-networks for forming efficient ensembles. The method is evaluated on ImageNet.

**Actions Required To Increase Overall Recommendation:**

In order to increase the overall recommendation, this paper would need to answer the open questions in Technical Quality And Correctness and clearly state the novelty.

**Clarity:**

This paper is rather hard to follow and I am confused about the actual contribution and novelty in this paper in contrast to OFA².

**Overall Review:**

Strengths:
The experiments show good results in finding architectures with high accuracy and low latency in the proposed setting.
This paper tackles an important research question.

Weaknesses:
There are some open questions (see Technical Quality And Correctness), which would when answered improve the overall assessment and novelty of the approach in comparison to other multi-objective search strategies.
A clear description of the strategy is missing.



**Potential Impact On The Field Of Automl:**

This paper discusses an important field of AutoML, but I find the way of the presentation hard to understand, and the novelty in comparison to OFA² is rather limited.

**Review Confidence:**

3: You are fairly confident in your assessment. It is possible that you did not understand some parts of the submission or that you are unfamiliar with some pieces of related work.

**Review Rating:**

3: Reject: For instance, a paper with technical flaws, weak impact, and/or weak evaluation.

**Review Summary:**

My main concern is that there are no comparisons with other methods except for OFA² and most importantly the contribution is not clear to me.

Additionally, the submission checklist is filled rather sparsely.


**Technical Quality And Correctness:**

There are some open questions regarding the technical quality of the proposed approach:

What is the search space?
How does the supernetwork look like? Since OFA is the basis for this work, more information on that is necessary.
Some information about NSGA-II is also helpful (in the appendix would be sufficient).

Lines 237-238:  I don’t understand this sentence: Why are all ensembles equal?

Line 247: “typical Pareto-front”:  I would be rather careful with this sentence, since this sounds like, the found ensembles are the Pareto front, and I don’t see the guarantee of more information about that.

What are the search times to find the architectures/ensembles?

What are the resulting accuracy/latency numbers in comparison to other baseline methods?

How does this method behave for other objectives and not only latency?

Line 297-298: “consistently outperform”. This paper only shows results in comparison to OFA², therefore this sentence is rather overstated.


Why was hard voting introduced in that detail, if not used in the paper?

---

> ### Author Response · Authors · 2023-05-06
> **Response to Reviewer tDcb (part 1)**
>
> Dear reviewer tDcb, thank you for your time and your valuable comments addressed to our work. We adjusted our paper taking your comments into consideration. Next, we copied some of your comments and added our reply straight after it:
>
> - [… the novelty in comparison to OFA² is rather limited.]
>
> The OFA³ proposal involves a cascade of three once-for-all mechanisms during the NAS: a single training step (provided by OFA), a single search step to populate the Pareto frontier (provided by OFA²), and a single selection step to compose the ensemble of efficient learning models, provided by OFA³. This is a remarkable achievement due to two main reasons: (1) OFA has split training and searching the neural architecture, thus making the search stage of negligible cost, when compared to the training of the super-network. Therefore, there is room for more elaborate search strategies, as performed by OFA² and OFA³, both being of a multi-objective nature and cascading. (2) Making a multi-objective selection of efficient components for the ensemble (being the main contribution of OFA³) is motivated by three main factors: (2.1) the guaranteed presence of distinct trade-offs among the candidate components provided by OFA², given that they populate an approximation of the Pareto frontier; (2.2) the assurance that they are independent models and can operate fully in parallel; (2.3) the possibility of automatically choosing just a subset of the efficient models produced by OFA² as components of the best ensemble. Those are the main motivation to support the gain in performance when compared with, for instance, a single model of the same size of the whole ensemble, which, to make matters worse, would not be implementable by resorting to independent fully parallelizable sub-networks. We have included this additional discussion at the concluding remarks of the paper.
>
> - [How does this method behave for other objectives and not only latency?]
>
> This is a relevant question that will be investigated as a further step of the research (we have included support to FLOPS in our experimental setup already). The OFA³ framework is immediately extensible to deal with additional conflicting objectives (joining latency and accuracy) and/or distinct conflicting objectives (replacing latency and/or accuracy). We have included this statement at the concluding remarks of the paper.
>
> - [A clear description of the strategy is missing.]
>
> Given that we have one additional page to prepare the final version of the paper, we have enriched the presentation of OFA³ strategy, at Section 1 (Introduction) and at Section 3 (Methodology).

---

> ### Author Response · Authors · 2023-05-06
> **Response to Reviewer tDcb (part 2)**
>
> Next, we copied some of your comments and added our reply straight after it:
>
> - [My main concern is that there are no comparisons with other methods except for OFA²]
>
> OFA performance has already been contrasted, in its original proposal, with the performance of a high number of relevant contenders, with a clear advantage both in terms of accuracy and computational burden. So, making a direct comparison solely with OFA and OFA² is a valid methodology here, because OFA was already properly positioned before relevant contenders in the literature.
>
> - [What is the search space?]
>
> Since we are dealing with an encoding of 100 binary genes, the search space of the OFA³ is 2^{100}. We also included this information on the text (lines 215-216).
>
> - [How does the supernetwork look like? Since OFA is the basis for this work, more information on that is necessary.]
>
> The OFA supernetwork is formed by 5 convolutional units placed in sequence. Each unit have 3 different parameters that can have their values changed: depth (number of layers) chosen from {2, 3, 4}, channel expansion ratio (number of channels) chosen from {3, 4, 6} and the convolution kernel size chosen from {3, 5, 7}. This gives OFA a search space of approximately 10^{19}. We did not include this information since it was already very well discussed in the papers of OFA and OFA², which are related to the architecture search. Our method uses the OFA and OFA² as the basis of our work. However we are tackling a different problem here, which is the selection of networks to form efficient ensembles, instead of architectures search. For the final version of the paper, we plan to include this information on the appendix, though.
>
> - [Lines 237-238: I don’t understand this sentence: Why are all ensembles equal?]
>
> The first population of ensembles is started with all genes set to one (meaning that all architectures from the pool of 100 neural networks participate in the ensembles). Therefore, all individuals of the first population have the same encoding and represent in fact the same ensemble. We have now included this information in the text to make it more clear: Sections 4.1 - Encoding (lines 226\~230), 5.1 - Summed latency (lines 252\~254) and 5.2 - Maximum latency (lines 281\~283).
>
> - [Line 247: “typical Pareto-front”: I would be rather careful with this sentence, since this sounds like, the found ensembles are the Pareto front, and I don’t see the guarantee of more information about that.]
>
> We changed this sentence in the text (lines 264-265).
>
> - [What are the search times to find the architectures/ensembles?]
> - [What are the resulting accuracy/latency numbers in comparison to other baseline methods?]
>
> We included a table of comparison between OFA³ (our method) and the baselines OFA and OFA². In this Table 1 there are information on latency, accuracy and search time (GPU hours).
>
> - [Additionally, the submission checklist is filled rather sparsely.]
>
> We refactored the submission checklist and now it is complete.

---

### Official Review · Reviewer_e426 · 2023-04-24

**Potential Impact On The Field Of Automl Rating:** 3
**Technical Quality And Correctness:** The approach seems to me correct.
**Technical Quality And Correctness Rating:** 4
**Clarity Rating:** 3
**Actions Required To Increase Overall Recommendation:** Please add a table of experiments.

**Summary Of Contributions:**

The authors develop OFA^3, a continuation of OFA^2.
OFA^2 is first trained to create a large network that during search time, given the accuracy and latency requirement, outputs the best model (subnetwork) it can find for the required latency and accuracy.
The authors use all the subnetworks that OFA^2 outputs, and then using evoulutionary algorithm (EA) they output the best ensemble of these subnetworks for the given latency-accuracy requirement.
Their EA is based on the NSGA-II algorithm. The DNA is a bit-string whose size matches the number of subnetworks output by the OFA^2 model.
They start with all bits set to 1. The mutation flips a bit at random with some probability p=1%, the crossover operator is the uniform crossover so every gene is chosen uniformaly at random from either of the parents, and the selection algorithm is based on non-dominated sorting and crowding distance metrics, as in NSGA-II.
The authors then experiment using two options: sum of latencies of the subneworks in the ensemble, and the maximum latency of the subnetworks in the ensemble. In the case of sum of latencies, they find out that it may be better to use a single architecture rather than an ensemble in most of the cases. For the maximum latency, the authors show that their results dominate any other configuration of the available sub-networks, taking accuracy and latency as the conflicting objective.


**Clarity:**

Strength:
The paper is very well written, the introduction of every topic is well explained and related work is discussed.

Weakness:
There is not even a single comparison table in the paper. Please add some comparison table, with numerical values. In my opition, it is important for the evaluation of the paper as well as a way to compare future work to your own work.

**Overall Review:**

Strengths
========
S1. An interesting approach to combine the subnetworks of OFA^2 into an ensemble.
S2. The approach is simple, the paper is well written and easy to follow, the figures are explanetory.

Weaknesses
==========
W1. There is not even a single comparison table in the paper. Please add some comparison table, with numerical values. In my opition, it is important for the evaluation of the paper as well as a way to compare future work to your own work.

W2. The evolutionary algorithm is pretty simple / straightforward, maybe consider several alternatives and show what is the best.

W3. Can you add ablation study.

Not listed as a weakness, but, the approach uses the subnetworks computed by OFA^2, and run an EA on the output of OFA^2. Can you find an approach that is more entangled with the training and search procedure of OFA^2, I find it interesting as well.


**Potential Impact On The Field Of Automl:**

A postprocessing phase to OFA^2 that uses EA to compose ensembles of subnetworks, thought the EA is relatively straightforward.
It seems to me as a nice incremental contribution, though not groundbreaking.


**Review Confidence:**

3: You are fairly confident in your assessment. It is possible that you did not understand some parts of the submission or that you are unfamiliar with some pieces of related work.

**Review Rating:**

6: Borderline Leaning Accept: Technically sound paper where reasons to accept outweigh reasons to reject. Please use sparingly.

**Review Summary:**

See the above strengths and weaknesses.

---

> ### Author Response · Authors · 2023-05-06
> **Response to Reviewer e426**
>
> Dear reviewer e426, thank you for your time and your valuable comments addressed to our work. We adjusted our paper taking your comments into consideration. Next, we copied some of your comments and added our reply straight after it:
>
> - [W1. There is not even a single comparison table in the paper. Please add some comparison table, with numerical values.]
>
> A table containing comparative results was added. However, it is worth mentioning that OFA performance has already been contrasted, in its original proposal, with the performance of a high number of relevant contenders, with a clear advantage both in terms of accuracy and computational burden. So, making a direct comparison solely with OFA and OFA² is a valid methodology here, because OFA was already properly positioned before relevant contenders in the literature.
>
> - [W2. The evolutionary algorithm is pretty simple / straightforward, maybe consider several alternatives and show what is the best.]
>
> We included support to other evolutionary algorithms as well. We realized experiments with the NSGA-II, SMS-EMOA and SPEA2. These experiments are provided as Jupyter notebooks as follows: [OFA² search](https://anon-github.automl.cc/r/once-for-all-3-89E3/jupyter-notebooks/ofa2.ipynb), [OFA³ (summed latency)](https://anon-github.automl.cc/r/once-for-all-3-89E3/jupyter-notebooks/ofa3-summed-latency-N1000.ipynb) and [OFA³ (maximum latency)](https://anon-github.automl.cc/r/once-for-all-3-89E3/jupyter-notebooks/ofa3-max-latency.ipynb). We also ran the experiments for 3 different random seeds each.
>
> - [W3. Can you add ablation study.]
>
> Our ablation study is related to three subjects: voting schemes (hard/soft voting), latency computation (summed/maximum) and now the evolutionary algorithm used during the search (NSGA-II, SMS-EMOA or SPEA2). We plan to extend the results on Table 1 to include each combination of these choices, since they are independent to each other, in our final version of the paper.
>
> - [Not listed as a weakness, but, the approach uses the subnetworks computed by OFA^2, and run an EA on the output of OFA^2. Can you find an approach that is more entangled with the training and search procedure of OFA^2, I find it interesting as well.]
>
> We provided several Jupyter notebooks ([GitHub repository](https://anon-github.automl.cc/r/once-for-all-3-89E3) --> jupyter-notebooks) on how to run each of these techniques (OFA, OFA² and OFA³). But your suggestion is valid, and we plan to add this support to the code of OFA³ as well.

---

### Meta-Review · Area_Chair_mSEC · 2023-05-10

**Recommendation:** Reject
**Confidence:** 4

**Metareview:**


The paper proposes an ensembling strategy on top of the OFA approach to bundle sub-networks. Reviewers agreed that this is an interesting approach. However, in its current state, the paper contains several shortcomings. First, the novelty is limited and the main approach is a rather straight-forward extension of previous work. Ensembling is well-known to outperform individual models and the proposed approach is based on basic majority voting / averaging of predictions. Second, several reviewers pointed out issues in the empirical evaluation of the method. For example there is no comparison to other methods than OFA2. Even tough OFA’s performance has been benchmarked in the original paper, the paper would be more rigorous if it contains it owns benchmarking and ideally confirms the reported results.

---

### Decision · Program_Chairs · 2023-05-15

Reject